# Real-Time Tension Distribution Design for Cable-Driven Parallel Robot

**Sheng Cao \* , Zhiwei Luo and Changqin Quan**

Graduate School of System Informatics, Kobe University, 1-1 Rokkodai-cho, Nada-ku, Kobe 657-8501, Japan
\* Correspondence: jasonsosen@gold.kobe-u.ac.jp

**Abstract:** In this study, we investigated dynamic control strategies for over-constrained cable-driven robots. In order to control a cable-driven robot, it is essential to address issues that arise from the restriction of cable tension, as well as to the redundancy issues that arise from an over-constrained cable-driven system. In contrast to previous research that required consideration of the relationship between tension constraints and computed control wrench in tension distribution problems, we developed a tension function that incorporates the hyperbolic tangent function, which allows tension to always satisfy tension constraints and eliminates the consideration of constraints at each step. The gradient descent method was applied to this tension function to determine an appropriate distribution of tension for the computed wrench. In order to manage tension distribution optimization for achieving objectives such as energy conservation, we provide a practical method to simultaneously realize the necessary wrench and the appropriate tension distribution. Compared with studies that focus on the complex analysis of the structure matrix to solve the tension distribution problem, the tension distribution issue is handled in a straightforward manner in our method, providing the solutions to other problems, such as discontinuity in the calculated wrench, and requirements of changing the cable's force level during movement. The simulation results and results of comparison with other methods show the effectiveness of the method.

**Keywords:** cable-driven robot; tension constraint; torque control; adaptive algorithm for Jacobian matrix

## 1. Introduction

A cable-driven parallel robot is a type of parallel robot that utilizes cables in place of the conventional rigid links to drive the end-effector of the robot. Within the framework of this system, the end-effector is linked to a number of cables that are driven by rotary motors. The cable-driven mechanism utilized by these types of robots is advantageous for a variety of reasons: it is lightweight, provides a large workspace, and is easy to transport and reconfigure.

Due to these advantages, cable-driven robots are well-suited for use in a wide variety of applications, including rehabilitation robots, camera systems, multiple cooperative cranes, and more.

Nevertheless, in order to properly design either the controller or the structure of such a robot, two aspects of the cable-driven robot system need to be taken into consideration. One is that the tension of the cable must be maintained at a positive value, as the cable can only pull the end-effector of the robot and cannot be used to push it. In addition, the tension of the cable ought to be limited at a maximum value in order to accommodate the saturation of the motor and the tensile strength of the cable.

The other is the requisite redundancy design implemented in such robot systems. Although there have been are various categories proposed for cable-driven robots based on the number of cables and the robot's degrees of freedom, in practice, cable-driven robots can be divided into fully constrained, over-constrained, and under-constrained robots, denoted as $m = n, m > n, m < n$, respectively, where $n$ denotes the degree of freedom of the robot and $m$ represents the number of cables.

In practice, analysis and control design are challenging due cable-driven robots' lack of pushing capabilities. In addition to requiring that the saturation problem be factored into the control strategy, this impacts the feasible workspace of the robot. In [1], the authors classified the SEW (static equilibrium workspace) as the set of poses attained statically considering the gravity. In the research of [2,3], the WCW (wrench closure workspace) is defined as the set of poses in which cable tensions can sustain an arbitrary external wrench. In other scenarios, such as [4,5], the workspace where cable tensions can produce any bounded wrench in the required set is referred to as the WFW (wrench feasible workspace).

In addition to the analysis of the feasible workspace of robots, many studies have concentrated on the control problem, that is, how to produce feasible tension to accomplish the control objective. The primary difficulty stems from the tension limits in combination with the issue of necessary redundancy in the over-constrained cable-robot configuration. In [6], an optimal tension distribution law for a computed PD control input from the viewpoint of workspace conditions was proposed. In [7], a feedback tracking control method based on the Control Lyapunov Function for cable suspended robots was presented. However, these approaches fail at other control aims, such as force control. In [8], an LP (Linear Programming) and QP (Quadratic Programming) solver was used to calculate the positive tension for a computed wrench. In [9], an LP solver for the optimal positive solution of the tension was able to provide rapid calculation. Verhoeven [10] developed an optimization algorithm for cable robots that minimizes the p-norm objective function, especially for high values of p. The LP and QP methods utilized in these investigations can achieve an optimal selection of positive tension distributions. However, the programming method used in these studies can cause a discontinuity in the cable tension, which renders the robot's control unstable. Furthermore, such programming methods can require an excessive amount of time to search for a feasible distribution in each step, making it difficult to realize the desired real-time calculation [11].

In addition to the programming-based tension distribution approach, there exist other studies on this issue that solve the problem by examining the relationship between the structural equation's solution space and the tension constraint's hypercube. Hassan [12,13] introduced the iterative Dykstra method, which computes tension distribution by locating the intersection between the solution space of the structural equation and the hypercube of the tension constraint. However, this method cannot compute force distributions that are continuous throughout a trajectory, and the rate of convergence is slow. In order to solve the problems of computation speed and continuous solution, Pott suggested a closed-form method in [14], followed by an improved version in [11]. Furthermore, the Barycentric force distribution method and its improved version, which can realize real-time calculation for cable-driven robots with two redundant cables, are provided in [15–17]. The puncture method [18] is another real-time approach that does not require a certain redundancy level. Although these algorithms calculate a continuous sequence of tension distributions, they only consider a certain minimum, maximum, or medium value, which may not be sufficient for certain applications, for example, those that require force control of the end-effector through stiffness control (where stiffness is affected by the tension in each cable). In addition, they only seek to solve the continuous problem by searching for solutions in the kernel space of the structure matrix, neglecting the possible discontinuity in the computed control wrench.

In this paper, we present a straightforward tracking control strategy for a cable-driven robot with a hyperbolic tangent function for force distribution. The tension is expressed here as a function with respect to a vector $L \in \mathbb{R}^m$ with a direction that is not specified. Utilizing the bounding attribute of the hyperbolic tangent function, this function satisfies tension requirements with user-selectable upper and lower bounds; $L$ is chosen in relation to the computed PID control input in the wrench space. In addition, we propose a method for selecting an optimal tension distribution in relation to the computed wrench in tension space in order to carry out the tension distribution optimally and thereby achieve certain objectives, such as the minimization or maximization of the cable's tension

vector norm. The proposed method may address the issue of potential discontinuity in the computed control wrench. This characteristic is highlighted in the comparison of our method with existing methods in the discussion section. Based on the proposed tension distribution method, we provide an algorithm that adaptively controls a cable-driven robot with physical parameters, such as its gravity center, that are unknown. Appropriate parameter selection in our tension distribution approach can manage adaptive tuning parameter-induced control input vibration, resulting in continuous tension.

The rest of our paper is organized as follows: first, we introduce the problem formulation in Section 2, followed by the proposed method for tension distribution to track robot control in Section 3. A version of this method with consideration of optimization is proposed in Section 4. In Section 5, we propose an adaptive tracking control law for cable-driven robots to account for uncertain parameter of the structure matrix of cable-driven robots. The efficiency of our strategy is demonstrated by the simulation results presented in Section 6. We provide a discussion of comparisons of our method and previous methods in Section 7. Lastly, we conclude the paper in Section 8.

## 2. Problem Formulation

### 2.1. Robot Kinematics Analysis

The general kinematics model of cable-driven parallel robots is shown in Figure 1.

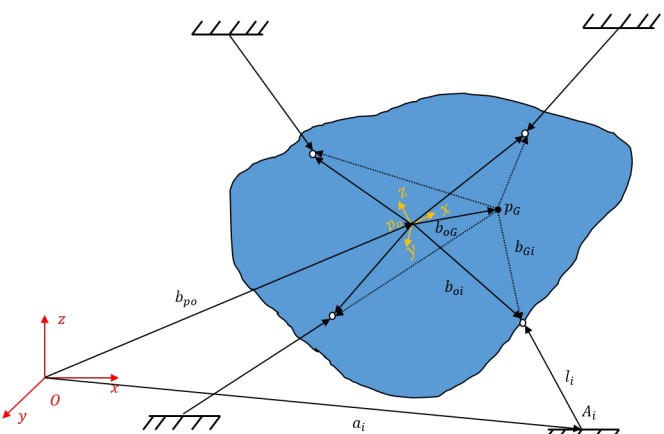

**Figure 1.** Model of cable-driven parallel robot.

A cable-driven robot consists of n cables and one end-effector. Each cable is driven by one motor. To explain how the robot moves, we can use the following two coordinate systems: first, an base coordinate system $\mathbb{C}_o = \{O_o, x_o, y_o, z_o\}$ fixed at one point at the workspace, and second, an end-effector coordinate system $\mathbb{C}_p = \{O_p, x_p, y_p, z_p\}$ attached at one point of the end-effector.

Moreover, in Figure 1, the position of the gravity center represented in $\mathbb{C}_o$ and $\mathbb{C}_p$ is denoted as $p_G$ and $^p p_G$, respectively; $b_{oG}$ denotes the vector from the selected origin of $\mathbb{C}_p$ with the unknown gravity center $p_G$ in $\mathbb{C}_o$, and $^p b_{oG}$ represents the same vector in $\mathbb{C}_p$. The location of the $i$th motor in the original coordinate system is denoted as $A_i$, with $i = 1, \ldots, n$ in $\mathbb{C}_o$, and the cable anchor point at the end-effector is represented as $B_i$ and $^p B_i$ in $\mathbb{C}_o$ and $\mathbb{C}_p$, respectively. Here, $b_{po}$ represents the vector from the origin of $\mathbb{C}_o$ to the origin of $\mathbb{C}_p$ in $\mathbb{C}_o$, and the rotation relation between the two coordinate systems is formulated by the rotation matrix $R$ shown below, which is associated with the three Euler angles $\phi, \theta, \psi$, respectively representing the roll, pitch, and yaw angles [15].

$$R = \begin{bmatrix} c_\psi c_\theta & c_\psi s_\theta s_\phi - s_\psi c_\phi & c_\psi s_\theta c_\phi + s_\psi s_\phi \\ s_\psi c_\theta & s_\psi s_\theta s_\phi + c_\psi c_\phi & s_\psi s_\theta c_\phi - c_\psi s_\phi \\ s_\theta & c_\theta s_\phi & c_\theta c_\phi \end{bmatrix} \tag{1}$$

Thus, the relationship between the time variation of cable length and the motion of the gravity center of the end-effector can be shown as follows:

$$\dot{l} = \bar{J}[\dot{p}_G{}^T \quad \Omega^T]^T \tag{2}$$

where $\dot{l} := [\dot{L}_1, \ldots, \dot{L}_m]^T \in \mathbb{R}^{m \times 6}$; here, $l$ represents the vector of the cable length and $L_i$ represents the cable length of the $i$-th cable. Here, $\bar{J}$ is defined as [15]:

$$\bar{J} := [\zeta_1 \quad, \ldots \quad, \zeta_m]^T \tag{3}$$

where $\zeta_i := [u_i^T \quad, (b_{Gi} \times u_i)^T]^T \in \mathbb{R}^{6 \times 1}$ for $i = 1, \ldots, n$, with $b_{Gi} := B_i - p_G$, and $u_i := l_i/L_i = (B_i - A_i)/L_i$ represents the normalized cable direction; furthermore, $l_i = L_i u_i$ represents the cable vector. Note that $b_{Gi}$ can be rewritten as $b_{Gi} = R(^p b_{oi} - {}^p b_{oG})$, where $^p b_{oG}$ is the position of the gravity center. The angular velocities can be related to the Euler angles as follows:

$$\Omega := \Theta \omega = \Theta[\dot{\phi} \quad \dot{\theta} \quad \dot{\psi}]^T \tag{4}$$

with $\omega = [\dot{\phi}, \dot{\theta}, \dot{\psi}]^T$ and the matrix $\Theta \in \mathbb{R}^{3 \times 3}$ is calculated as follows:

$$\Theta := \begin{bmatrix} 1 & 0 & -s_\theta \\ 0 & c_\phi & c_\theta s_\phi \\ 0 & -s_\phi & c_\theta c_\phi \end{bmatrix} \tag{5}$$

### 2.2. Robot Dynamics Analysis

By introducing the generalized coordinates vector $q = [p_G^T, \phi^T, \theta^T, \psi^T]$ for the pose of the end-effector, the robot dynamics can be expressed as follows [15]:

$$M(q)\ddot{q} + C(q, \dot{q})\dot{q} + G(q) = A^T f \tag{6}$$

where

$$M(q) = \begin{bmatrix} m I_{3 \times 3} & 0_{3 \times 3} \\ 0_{3 \times 3} & \Theta^T I_p \Theta \end{bmatrix} \tag{7}$$

$$C(q, \dot{q}) = \begin{bmatrix} 0_{3 \times 3} & 0_{3 \times 3} \\ 0_{3 \times 3} & \Theta^T I_p \dot{\Theta} + \Theta^T (\Omega)_\times (I_p \Theta) \end{bmatrix} \tag{8}$$

$$G(q) = \begin{bmatrix} 0 \\ 0 \\ -mg \\ 0_{3 \times 1} \end{bmatrix} \tag{9}$$

$I_p$ denotes the inertia tensor of the end-effector about point $P_G$ in $\mathbb{C}_o$, and $A \in \mathbb{R}^{m \times 6}$ is the Jacobian matrix, which is defined as follows:

$$A = -\bar{J}\gamma \tag{10}$$

with $\gamma := \begin{bmatrix} I_{3 \times 3} & 0_{3 \times 3} \\ 0_{3 \times 3} & \Theta \end{bmatrix}$. The condition that applies is that the tension of the $i$th cable must be $f_i \in [f_{\min}, f_{\max}]$, where $f_{\min}$ and $f_{\max}$ denote the lower and upper boundaries of one cable, respectively, and $f_{\min}$ must be positive. Furthermore, as previously noted, because the robot's control wrench must always satisfy the wrench closure constraint, the number of cables $n$ and the degree of freedom of the robot's system $n$ must always satisfy the equation $m \geqslant n + 1$, which implies that a cable-driven system must always have redundancy. Finding a solution to the tension distribution problem requires taking into account both the tension constraint and the redundancy of the robots at the same time.

Previous studies have employed optimization techniques, such as linear programming, to determine a feasible control input that satisfies the tension constraint. However, these

approaches have limitations, such as a discontinuous solution or a complicated computation procedure. Real-time applications necessitate a technique that can both make the tension distribution problem continuous and reduce the amount of time spent computing the solution.

Here, we present a method for determining the appropriate control input for a cable-driven robot that performs tension selection based on a tension function containing the hyperbolic tangent function. Because of the boundary constraint imposed by the hyperbolic tangent function, the proposed tension function achieves the desired tension limits. Adjusting the parameters within this tension function produces a change in tension that is suitable for the desired wrench. Compared to conventional optimization approaches, this strategy can determine the optimal tension distribution more rapidly and effectively while preserving the continuity of the tension's variation.

### 3. Tracking Control of Cable-Driven Robot with Hyperbolic Tangent Function-Based Tension Distribution

*3.1. Hyperbolic Tangent Function-Based Tension Distribution Method of Cable-Driven Robot with Known Jacobian Matrix*

We can select the desired control input of the robot system as follows in order to obtain the tracking control of a cable-driven robot [15]:

$$A^T f = \tau_d = M\ddot{q}_r + C\dot{q} + G \tag{11}$$

where $q_r$, the two-order time derivative of which is $\ddot{q}_r = k_p(q_d - q) + k_d(\dot{q}_d - \dot{q}) + k_i \int_0^t (q_d - q)\mathtt{dt}$, represents the reference trajectory. From Equation (11), it is possible to achieve PID tracking control of a cable-driven robot with nonlinear compensation by properly adjusting the tension $f$ in accordance with the unique Jacobian matrix $J$ and desired wrench $\tau_d$.

We suggest a strategy for the distribution of continuous tension that utilizes a simple computation procedure. The value $f$ is chosen based on the function defined below:

$$f = f_{\mathtt{min}}\mathbb{1} + \lambda[\mathbb{1} + \mathtt{Tanh}(L)] \tag{12}$$

where $\mathbb{1} \in \mathbb{R}^n$ denotes a vector in which all elements are 1, $L = [L_1, \ldots, L_m]^f \in \mathbb{R}^n$ represents a vector with an unknown value and that must be adjusted in accordance with the desired control input in wrench space, and $\mathtt{Tanh}(L) = [\mathtt{tanh}(L_1), \ldots, \mathtt{tanh}(L_m)]^T$ and $\lambda$ represent a positive scalar related to tension constraints, calculated as $\lambda = \frac{f_{\mathtt{max}} - f_{\mathtt{min}}}{2}$. Consequently, considering the hyperbolic tangent function's boundary condition ($-1 \leqslant \mathtt{tanh}x \leqslant 1$, with $x$ denoting any scalar), any selection of $L$ can satisfy the tension constraints ($f_i \in [f_{\mathtt{min}}, f_{\mathtt{max}}]$).

Then, using Equations (11) and (12), we can obtain

$$A^T(f_{\mathtt{min}}\mathbb{1} + \lambda[\mathbb{1} + \mathtt{Tanh}(L)]) = \tau_d \tag{13}$$

This equation can be expressed as follows:

$$A^T\mathtt{Tanh}(L) = \bar{\tau} \tag{14}$$

where $\bar{\tau} = \frac{\tau_d - (f_{\mathtt{min}} + \lambda)J\mathbb{1}}{\lambda}$. As a result, we can determine $L$ to realize $\bar{\tau}$ and then further realize the needed control input $\tau_d$.

An error function can be constructed as $E = \frac{1}{2}e^T e$, where the error $e$ is defined as $e = A^T\mathtt{Tanh}(\hat{L}) - \bar{\tau}$ and $\hat{L}$ denotes the estimation of $L$.

$E$ can be minimized by appropriately tuning $\hat{L}$ using the gradient descent method. Variations of $\hat{L}$ can be presented as

$$\dot{\hat{L}} = -\alpha \frac{\partial E}{\partial \hat{L}} = -\alpha(\boldsymbol{U}_{m \times m} - \mathtt{Tanh2}(\hat{L}))^T A e \tag{15}$$

where $U_{m \times m}$ represents the unit matrix in $\mathbb{R}^{m \times m}$ space and $\alpha$ is a positive scalar; $\texttt{Tanh2}(\hat{L}) \in \mathbb{R}^{m \times m}$ represents a diagonal matrix function defined as follows:

$$\texttt{Tanh2}(\hat{L}) := \begin{bmatrix} \texttt{tanh}(L_1)^2 & \cdots & 0 \\ \vdots & \ddots & \vdots \\ 0 & \cdots & \texttt{tanh}(L_m)^2 \end{bmatrix} \tag{16}$$

*3.2. $\alpha$ Selection Condition for Convergence of the Error e*

By applying the updated law in Equation (15), the error $e$ might be minimized when the time derivative of the desired wrench $\tau_d$ is zero or very small. However, if $\tau_d$ varies over time, updating $\hat{L}$ to reflect this variation requires additional analysis and the choice of a suitable $\alpha$.

Here, the derivative of $E$ can be formulated as follows:

$$\dot{E} = e^T \dot{e} = e^T(\dot{A}^T \texttt{Tanh}(\hat{L}) + A^T \frac{\mathrm{d}}{\mathrm{dt}}(\texttt{Tanh}(\hat{L})) - \dot{\bar{\tau}}) \tag{17}$$

As $\bar{\tau} = \frac{\tau_d - (f_{\min} + \lambda)J\mathbb{1}}{\lambda}$, $\dot{\bar{\tau}}$ can be shown as follows:

$$\dot{\bar{\tau}} = \frac{\dot{\tau}_d - (f_{\min} + \lambda)\dot{A}^T\mathbb{1}}{\lambda} \tag{18}$$

We can rewrite Equation (17) as follows:

$$\begin{aligned} \dot{E} = &\, e^T(\dot{A}^T \texttt{Tanh}(\hat{L}) + A^T \frac{\mathrm{d}}{\mathrm{dt}}(\texttt{Tanh}(\hat{L})) - \frac{1}{\lambda}\dot{\tau}_d \\ &+ \frac{1}{\lambda}(f_{\min} + \lambda)\dot{A}^T\mathbb{1}) \end{aligned} \tag{19}$$

where

$$\begin{aligned} \dot{\tau}_d = &\, \dot{M}(-k_p(q - q_d) - k_d\dot{q}) + C\ddot{q} + \dot{C}\dot{q} + \dot{G} \\ &+ M(-k_p(\dot{q} - \dot{q}_d) - k_d\ddot{q}) \\ &\frac{\mathrm{d}}{\mathrm{dt}}(\texttt{Tanh}(\hat{L})) = (U_{m \times m} - \texttt{Tanh2}(\hat{L}))\dot{\hat{L}} \end{aligned}$$

Furthermore, by incorporating Equation (15), it can be concluded that

$$\dot{E} = e^T z - \alpha d \tag{20}$$

where

$$\begin{aligned} z = &\, \dot{A}^T \texttt{Tanh}(\hat{L}) + \frac{1}{\lambda}((f_{\min} + \lambda)\dot{A}^T\mathbb{1}) - \frac{1}{\lambda}(C\ddot{q} + \dot{G} + \dot{C}\dot{q}) \\ &- \frac{1}{\lambda}(\dot{M}(-k_p(q - q_d) - k_d\dot{q}) + M(-k_p(\dot{q} - \dot{q}_d) - k_d\ddot{q})) \end{aligned} \tag{21}$$

$$d = e^T A^T H_{Tan} H_{Tan}^T A e \tag{22}$$

$$H_{Tan} = U_{m \times m} - \texttt{Tanh2}(\hat{L}) \tag{23}$$

$\ddot{q}$ can be calculated using $\ddot{q} = M^{-1}(A^T f - C\dot{q} - G)$. Thus, utilizing the dynamic and kinematic model of the robot and the sensor-measured angular position and velocities, it is possible to determine $z$.

From Equation (20), due to the fact that $d$ is a non-negative value, we can select an appropriate $\alpha$ to make $\dot{E}$ negative. Here, the value of $d$ is crucial for selecting $\alpha$.

**Remark 1.** *We do not need to update $\hat{L}$ and $\alpha$ if $d$ is close to zero.*

**Proof.** When $d$ decreases in value, each element of vector $H_{Tan}^T A e$ in tension space becomes small. It is evident that there may exist two possible explanations when $d$ is close to zero,

as each element of this vector is calculated using two variables, namely, the corresponding element of vector $Ae$ and the diagonal element of matrix $H_{Tan}$.

On the one hand, when the control wrench tracking error $e$ is decreased to zero and the desired control wrench is achieved, $d$ becomes zero.

On the other hand, when $d$ is close to zero, it is possible that diagonal elements of $H_{Tan}$ may be close to zero as well, as the corresponding elements of $Ae$ are not zero. Due to the fact that $H_{Tan}$ is a diagonal matrix and each diagonal element of $H_{Tan}$ is calculated as $1 - (\mathtt{tanh}(L_i))^2$, where $\mathtt{tanh}()$ has the upper and lower boundaries as $1$ and $-1$, respectively, each element of $H_{Tan}$ becomes small. This indicates that each cable's force is sufficiently close to the limitation boundary. The tension of the cable is actually unable to maintain the appropriate tension for realizing the desired wrench.

Therefore, in each of these circumstances there is no need to update $\hat{L}$ any further. $\square$

If $d$ is large enough, $\alpha$ can be selected provided the following conditions are met:

$$\alpha \begin{cases} \geqslant 0 & (\bar{E}_r > 0) \\ \geqslant \frac{-\bar{E}_r}{d} & (d > \delta, \bar{E}_r < 0) \end{cases}$$

where $\delta$ denotes a tiny positive value.

Here, $\bar{E}_r$ is defined as

$$\bar{E}_r := -e^T z - \beta e^T e \tag{24}$$

where $\beta$ is a positive scalar.

By following the condition of $\alpha$, we can obtain

$$-\alpha d \leqslant \bar{E}_r \tag{25}$$

Then, by substituting Equation (25) into Equation (20) and using Equation (24), we obtain

$$\dot{E} \leqslant -\beta e^T e \tag{26}$$

which indicates that the desired control wrench is tracked steadily.

## 4. Optimization of Cable Distribution

### 4.1. Optimization Algorithm

Although the previous section included a useful method for computing the tension distribution of the control wrench, it remains essential to select the proper tension distribution in an optimal and effective manner, and a method that provides a minimum norm for energy reservation is needed.

Notably, after utilizing the proposed tension function, we can easily achieve an optimal tension distribution by selecting or minimizing the cost function as $\mathtt{min}\ C(f)$ without tension constraints.

For instance, when choosing a tension with the lowest norm value in a tension space, we can choose $f$ as follows:

$$f_o = A^{+T} \tau_d \tag{27}$$

Nevertheless, $A^{+T} \tau_d$ is sometimes not a viable alternative, as it is possible that the tension requirement may not be satisfied. Consequently, a feasible $f$ that can track $A^{+T} \tau_d$ as closely as possible, meet the tension condition, and realize the desired control wrench is required.

In order to achieve this goal, we can augment the tension function as follows:

$$f = f_{\mathtt{min}} \mathbb{1} + \lambda [\mathbb{1} + \mathtt{Tanh}(\bar{S} L_a)] \tag{28}$$

with the use of $L$ in Equation (12):

$$L = \bar{S} L_a \tag{29}$$

where $\bar{S}$ represent a diagonal matrix, $\bar{S} := \begin{bmatrix} S_1 & \cdots & 0 \\ \vdots & \ddots & \vdots \\ 0 & \cdots & S_n \end{bmatrix}$.

Notably, $\bar{S}L_a$ can be computed as follows:

$$\bar{S}L_a = \bar{L}_a S \tag{30}$$

where $\bar{L}_a$ denotes the matrix $\bar{L}_a := \begin{bmatrix} L_{a_1} & \cdots & 0 \\ \vdots & \ddots & \vdots \\ 0 & \cdots & L_{a_n} \end{bmatrix}$ and the vector $S$ can be represented as

$S = [S_1, \ldots, S_n]^T$. Then, Equation (28) can be shown as $f = f_{\min}\mathbb{1} + \lambda[\mathbb{1} + \text{Tanh}(\bar{L}_a S)]$.

The error between the tension function and desired tension distribution with a minimum norm can be represented as follows:

$$e_2 = \hat{f} - f_o \tag{31}$$

and the desired control wrench tracking error becomes

$$e_1 = A^T \text{Tanh}(\hat{\bar{S}}\hat{L}_a) - \bar{\tau} \tag{32}$$

where $\hat{L}_a$ represents the estimation of $L_a$.

Here, the objective is to simultaneously minimize $e_1$ and $e_2$ in order to achieve the desired wrench and optimal tension distribution solution. We define an error function as $E_2 = e_2^T e_2$ and $E_1 = e_1^T e_1$ to represent the error of tracking the desired wrench and tracking the optimal tension distribution, respectively. In order to simultaneously minimize $E_1$ and $E_2$, based on the gradient descent method, the updated law of the estimation of $\hat{L}_a$ and $\hat{S}$ can be formulated as follows:

$$\dot{\hat{L}}_a = -\alpha_1 \frac{\partial E_1}{\partial \hat{L}_a} = -\alpha_1 \hat{S}^T (\boldsymbol{U}_{m \times m} - \text{Tanh2}(\hat{S}\hat{L}_a))^T A e_1 \tag{33}$$

and

$$\dot{\hat{S}} = -\alpha_2 \frac{\partial E_2}{\partial \hat{S}} = -\alpha_2 \lambda \hat{\bar{L}}_a^T (\boldsymbol{U}_{m \times m} - \text{Tanh2}(\hat{\bar{L}}_a \hat{S}))^T e_2 \tag{34}$$

where $\hat{\bar{L}}_a, \hat{L}_a, \hat{\bar{S}}, \hat{S}$ represent the estimation of $\bar{L}_a, L_a, \bar{S}, S$.

Notably, although the updating law chosen in Equations (33) and (34) minimizes $E_1$ and $E_2$, variations in either of these processes have repercussions on the other.

When the ideal tension $f_0$ satisfies all tension constraints, this tension becomes one of the tension solutions that minimize $E_1$ to zero. In this case, a decrease in $E_2$ facilitates decreases in $E_1$.

However, when the ideal tension $f_0$ is located outside of the tension constraints, reductions in $E_1$ and $E_2$ can actually lead to conflict, especially considering that updating $\hat{S}$ may cause a divergence in tension from the desired wrench. In detail, as seen in Figure 2, $f_0$ denotes the optimal tension selection out of the feasible tension space. The dashed line passing through $f_m$, $f_0$, and $\tau_d$ denotes the set of all the tension solutions corresponding to the desired $\tau_d$; the blue part of this dashed line in the tension space denotes the feasible tension solutions, $f_m$ denotes the feasible tension solution closest to $f_0$, and $f_p$ represents the solution that is nearest to $f_0$ in tension space. If we fix $\hat{L}_a$ and only update $\hat{S}$ following Equation (34), the tension value converges to $f_p$, reducing $E_2$ to the minimum. Moreover, even when we update the original fixed $\hat{L}_a$ with Equation (33), when $\hat{S}$ changes the value of the tension may deviate from the feasible tension solutions related to the desired control wrench.

Thus, in order to manage this situation, tracking of the desired wrench with an appropriate $\alpha_1$ is required to ensure that tensions are only selected from the points on

the blue part of the dashed line; by updating the law of Equation (34), $E_2$ becomes the minimum value on the blue part of the dashed line.

The total calculation and control process is shown in Algorithm 1.

To guarantee the convergence of $E_1$, in the following section we examine its dynamic change and define the tunable range of $\alpha_1$.

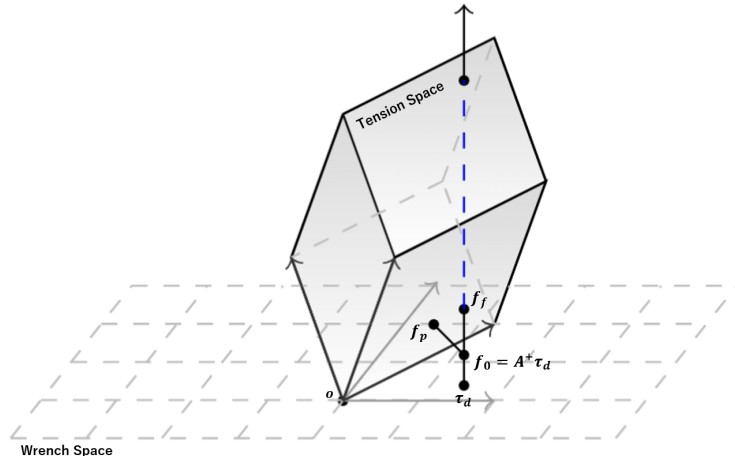

**Figure 2.** Relation between wrench space and tension space.

---

**Algorithm 1** tanh(:)-based Optimal Tension Distribution

---

**Require:** $q, \dot{q}, q_d, \hat{L}_{a0}, \hat{S}_0, \lambda$
**Ensure:** $f$
    *LOOP Process*
1: **for** $t = 0$ to $t_f$ **do**
2:    Calculate $\tau_d$ following Equation (11).
3:    Calculate the Updating law of Equations (33) and (34).
4:    Update the integral of $\hat{L}_a, \hat{S}$.
5:    Calculate the tension following Equation (28) and use the calculation result to control the robot.
6: **end for**

---

*4.2. Analysis for Tracking the Stability of the Desired Control Wrench*

Similar to the derivation of Equation (19), the derivative of $E_1$ can be calculated as follows:

$$\dot{E}_1 = e_1^T \dot{e}_1 = e_1^T (\dot{A}^T \text{Tanh}(\hat{L}) + A^T \frac{\text{d}}{\text{dt}}(\text{Tanh}(\hat{L})) - \dot{\tau}) \tag{35}$$

In order to examine the tracking stability of the desired control wrench, it is necessary to determine the variation of $\frac{\text{d}}{\text{dt}}(\text{Tanh}(\hat{L}))$ and $\dot{\tau}_d$. With the updating law selected as shown in Equations (33) and (34), $\frac{\text{d}}{\text{dt}}(\text{Tanh}(\hat{L}))$ becomes

$$\begin{aligned}
\frac{\text{d}}{\text{dt}}(\text{Tanh}(\hat{L})) &= (\boldsymbol{U}_{m \times m} - \text{Tanh2}(\hat{L}))\dot{\hat{L}} \\
&= (\boldsymbol{U}_{m \times m} - \text{Tanh2}(\hat{L}))(\hat{S}\dot{\hat{L}}_a + \hat{L}_a\dot{\hat{S}}) \\
&= -\alpha_1 H_{Tan}\hat{S}\hat{S}^T H_{Tan}^T A e_1 - \alpha_2 \lambda H_{Tan}\hat{L}_a\hat{L}_a^T H_{Tan}^T e_2
\end{aligned} \tag{36}$$

Equation (35) can be written as

$$\dot{E}_1 = e_1^T \bar{z} - \alpha_1 \bar{d} \tag{37}$$

where $\bar{z} = z - \alpha_2 \lambda H_{Tan}\hat{L}_a\hat{L}_a^T H_{Tan}^T e_2$ and $\bar{d} = e_1^T A^T H_{Tan}\hat{S}\hat{S}^T H_{Tan}^T A e_1$.

It has been shown that the term $-\alpha_2 \lambda H_{Tan} \hat{\bar{L}}_a \hat{\bar{L}}_a^T H_{Tan}^T e_2$ generated by the updating law of $\hat{S}$ has an effect on the tracking stability of the desired wrench. Here, we select $\alpha_1$ as follows to ensure consistent tracking of the desired wrench:

$$
\alpha_1 \begin{cases} \geqslant 0 & (\bar{\bar{E}}_r > 0) \\ \geqslant \dfrac{-\bar{\bar{E}}_r}{\bar{d}} & (\bar{d} > \delta, \bar{\bar{E}}_r < 0) \end{cases}
$$

where $\bar{\bar{E}}_r$ is defined as $\bar{\bar{E}}_r := -e_1^T \bar{z} - \beta e_1^T e_1$. As a result, we obtain

$$
\dot{E}_1 \leqslant -\beta e_1^T e_1 \tag{38}
$$

which indicates that $E_1$ continues to decrease with the suggested selection law for $\alpha_1$. Here, because the selection of $\alpha_1$ takes into account the effect of $\bar{\bar{E}}_r$, which includes the effect caused by the variation of $\hat{S}$, it can be seen that $A^T f$ is able to consistently track the desired wrench even when $\hat{S}$ varies. Due to the updating of $\hat{S}$, $E_2$ is minimized and the tension $\hat{f}$ finally reaches $f_m$, which is close to the desired $f_0$ illustrated in the feasible tension solutions with respect to $\tau_d$ (for example, $f_m$ in Figure 2).

*4.3. Analysis of Trajectory Tracking Control*

If the desired control of the wrench can be achieved, the robot's acceleration $\ddot{q}$, velocity $\dot{q}$, and position $q$ can satisfy the equation provided below due to the existence of the nonlinear compensated control wrench:

$$
\ddot{q} + K_d \dot{q} + K_p(q - q_d) + K_i \int_0^t (q - q_d) \mathtt{dt} = 0 \tag{39}
$$

which indicates that $q$ is well able to track the desired position $q_d$. However, because of the existence of the tracking error of the desired wrench, Equation (39) becomes

$$
\ddot{q}' + K_d \dot{q}' + K_p(q' - q_d) + K_i \int_0^t (q' - q_d) \mathtt{dt} = M^{-1}(A^T f - \tau_d) \tag{40}
$$

where $q'$ represents the novel system state.

By comparing Equation (39) to (40), an auxiliary error system can be constructed as follows:

$$
\ddot{q}_e + K_d \dot{q}_e + K_p q_e + K_i \int_0^t q_e \mathtt{dt} = d_e \tag{41}
$$

where $q_e := q' - q$.

Here, during the procedure in which $A^T f$ tracks $\tau_d$, the term $d_e = A^T f - \tau_d$ can be regarded as a disturbance that may cause a steady-state error in the robot's tracking control. In this study, the presence of the integral term $K_i \int_0^t (q - q_d) \mathtt{dt}$ in the nonlinear compensated PID controller can help to reduce the steady-state tracking error caused by this disturbance.

## 5. Tracking Control of Cable-Driven Robot Considering Model Uncertainties in Jacobian Matrix

*5.1. Analysis of Model Uncertainties' Impact on the Jacobian Matrix*

Considering the model uncertainties, Equation (15) becomes

$$
\dot{\hat{L}}' = -\alpha \frac{\partial E}{\partial \hat{L}'} = -\alpha (\boldsymbol{U}_{m \times m} - \mathtt{Tanh2}(\hat{L}'))^T \hat{A} e' \tag{42}
$$

where $\hat{A}^T$ represents the estimation of $A^T$, $\hat{L}', e'$ represent the new estimation of $L$ and the error $e$ using $\hat{A}^T$, respectively, $e'$ can be calculated with $e' = \hat{A}^T \mathtt{tanh}(\hat{L}') - \hat{\bar{\tau}}$, and $\hat{\bar{\tau}} = \frac{\tau_d - (f_{\mathtt{min}} + \lambda)\hat{A}^T \mathbb{1}}{\lambda}$.

Simultaneously, the robot's dynamics, as shown in Equation (6), are as follows:

$$M(q)\ddot{q} + C(q, \dot{q})\dot{q} + G(q) = A^T \hat{f} \tag{43}$$

Equation (43) can be rewritten as follows in order to analyze the effect of the estimation error on the robot's tracking behavior:

$$M(q)\ddot{q} + C(q, \dot{q})\dot{q} + G(q) = \tau_d - \tau_d + A^T \hat{f} = \tau_d + \tau_e \tag{44}$$

with $\tau_e = A^T \hat{f} - \tau_d$

Equation (41) becomes

$$\ddot{q}_e + K_d \dot{q}_e + K_p q_e + K_i \int_0^t q_e \mathtt{dt} = M^{-1}(A^T \hat{f} - \tau_d) \tag{45}$$

Considering that $q_e$ and $\dot{q}_e$ are both measurable, we suggest that an adaptive control law can be utilized to minimize the model errors. Furthermore, the previous equation can be rewritten as follows:

$$\ddot{q}_e + K_d \dot{q}_e + K_p q_e + K_i \int_0^t q_e \mathtt{dt} = M^{-1}(A^T \hat{f} - \hat{A}^T \hat{f} + \hat{A}^T \hat{f} - \tau_d) = \tau_{e_1} + \tau_{e_2} \tag{46}$$

where the $\tau_{e_1} = M^{-1}(A^T \hat{f} - \hat{A}^T \hat{f})$ denotes the error exerted by model uncertainties and $\tau_{e_2} = M^{-1}(\hat{A}^T \hat{f} - \tau_d)$ denotes the error between the desired wrench and approximated wrench.

Here, we can rewrite the term $M^{-1} A^T \hat{f}$ as $M^{-1} A^T \hat{f} = Ya$, where $Y$ denotes a regression matrix and $a$ represents the uncertain parameters in the Jacobian matrix. Then, $M^{-1} \hat{A}^T \hat{f}$ can be represented as follows:

$$M^{-1} \hat{A}^T \hat{f} = Y\hat{a} \tag{47}$$

where $\hat{a}$ denotes the estimation of $a$.

Equation (46) then becomes

$$\ddot{q}_e + K_d \dot{q}_e + K_p q_e + K_i \int_0^t q_e \mathtt{dt} = Y\tilde{a} + \lambda M^{-1} e' \tag{48}$$

According to Equation (48), $q_e$ is affected by both $Y\tilde{a}$ and $M^{-1}\lambda e'$. With the existence of $M^{-1}\lambda e'$, it is challenging to find a suitable time-changing law of $\hat{a}$ to accomplish the adaptive control law based on Equation (48). In this study, we employ a fault detection and isolation technique to build an auxiliary system with a state solely affected by $Y\tilde{a}$. We then propose an adaptive control law based on this novel system.

### 5.2. Fault Detection and Isolation-Based Adaptive Algorithm

First, as in [19,20], we can consider the dynamics of the generalized momentum of the auxiliary system in Equation (41), as follows:

$$p = \dot{q}_e \tag{49}$$

By differentiating Equation (49), we obtain

$$\begin{aligned} \dot{p} &= -(K_d \dot{q}_e + K_p q_e + K_i \int_0^t q_e \mathtt{dt}) + M^{-1}(A^T \hat{f} - \tau_d) \\ &= \tau_r + M^{-1} A^T \hat{f} \end{aligned} \tag{50}$$

For notational brevity, vector $\tau_r = -(K_d \dot{q}_e + K_p q_e + K_i \int_0^t q_e \mathtt{dt}) - M^{-1}\tau_d$ is introduced.

Based on $\tau_r$, we can construct a residual generator as follows:

$$r(t) = \Gamma[p(t) - \int_0^t (\tau_r + r(s))ds] \tag{51}$$

where $\Gamma$ denotes a positive scalar and the initial value of $r$ satisfying $r(0) = 0$. $r(t)$ is calculable as $p$ in $r(t)$, and can be calculated using $\dot{q} - \dot{q}'$, where $\dot{q}$ can be obtained using Equation (39) and $\dot{q}'$ can be measured by sensors.

By differentiating Equation (51) with respect to time and considering Equation (50), we can obtain

$$\dot{r} + \Gamma r = \Gamma M^{-1} A^T \hat{f} = \Gamma Y a \tag{52}$$

From this equation, it can be seen that the state $r$ of this constructed system is only affected by the term $\Gamma Y a$. If we introduce an estimation method using $\hat{a}$ instead of $a$, we obtain

$$\dot{\hat{r}} + \Gamma \hat{r} = \Gamma Y \hat{a} \tag{53}$$

where $\hat{r}$ represents new system states after using $\hat{a}$. By comparing the error between these two systems (Equations (52) and (53)), we can obtain the following error system:

$$\dot{\tilde{r}} + \Gamma \tilde{r} = \Gamma Y \tilde{a} \tag{54}$$

where $\tilde{r} = \hat{r} - r$ and $\tilde{a} = \hat{a} - a$. By appropriately adjusting $\hat{a}$, $\tilde{a}$ and $\tilde{r}$ can converge to zero and system Equation (54) becomes stable. The stability of this error system can be analyzed using a Lyapunov theory-based stability analysis, with the Lyapunov candidate selected as follows:

$$V = \frac{1}{2}\tilde{r}^T \tilde{r} + \frac{1}{2}\tilde{a}^T \tilde{a} \tag{55}$$

Therefore, the time derivative of $V$ can be calculated as follows:

$$\begin{aligned} \dot{V} &= \tilde{r}^T \dot{\tilde{r}} + \tilde{a}^T \dot{\tilde{a}} \\ &= -\tilde{r}^T \Gamma \tilde{r} + \tilde{r}^T \Gamma Y \tilde{a} + \tilde{a}^T \dot{\hat{a}} \end{aligned} \tag{56}$$

Thus, if we designate the time change of $\hat{a}$ as

$$\dot{\hat{a}} = -\Gamma Y^T \tilde{r} \tag{57}$$

we obtain

$$\dot{V} = -\tilde{r}^T \Gamma \tilde{r} \leq 0 \tag{58}$$

which means that the system of Equation (54) is stable. Here, $\tilde{r}$ converges to 0, which denotes that the error term $Y\tilde{a}$ converges to zero. With the dynamics in Equation (48), the tuning law of Equation (42), and the existence of the integral term in the nonlinear compensated PID controller, $q_e$ and $\dot{q}_e$ converge to zero, and $q'$ finally converges to the desired position $q_d$.

Here, in Equation (57), $\tilde{r}$ can be calculated using $r - \hat{r}$, where $\hat{r}$ can be obtained using Equation (53) and $r$ can be computed with Equation (51).

## 6. Simulation Studies

In order to verify the efficacy of our strategy, a simulation study of a cable-driven robot with six degrees of freedom and eight cables was conducted. This robot's geometric structure and configuration model are depicted in Figure 3.

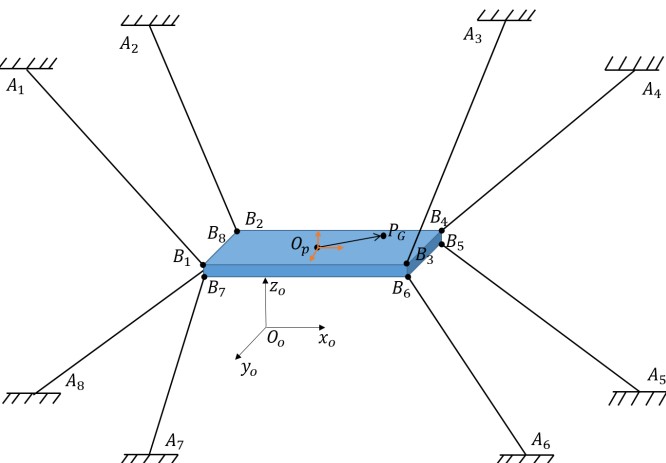

**Figure 3.** Cable-driven parallel robot with six degrees of freedom and eight cables.

The generalized robot's state can be represented as follows: $[x, y, z, \phi, \theta, \psi]^T$. The position of the eight motors in the original coordinate system is set as

$$A_1 = \begin{bmatrix} 1.5 \\ -1.5 \\ 2 \end{bmatrix}, A_2 = \begin{bmatrix} 1.5 \\ 1.5 \\ 2 \end{bmatrix}, A_3 = \begin{bmatrix} -1.5 \\ 1.5 \\ 2 \end{bmatrix}, A_4 = \begin{bmatrix} -1.5 \\ -1.5 \\ 2 \end{bmatrix},$$
$$A_5 = \begin{bmatrix} 1.5 \\ -1.5 \\ -2 \end{bmatrix}, A_6 = \begin{bmatrix} 1.5 \\ 1.5 \\ -2 \end{bmatrix}, A_7 = \begin{bmatrix} -1.5 \\ 1.5 \\ -2 \end{bmatrix}, A_8 = \begin{bmatrix} -1.5 \\ -1.5 \\ -2 \end{bmatrix} \tag{59}$$

with the angular parameters $(\phi, \theta, \psi)$ all set to zero. The tension constraint is set as $f_{\min} = 0$ N, $f_{\max} = 20$ N.

This robot's dynamics can be shown as follows:

$$\begin{bmatrix} m_{3\times3} & 0 \\ 0 & I_{3\times3} \end{bmatrix} \ddot{q} + G = A^T f \tag{60}$$

where $m_{3\times3}$ and $I_{3\times3}$ represent a diagonal matrix with diagonal elements set as the mass $m$ and the inertia $I$ of the end-effector, respectively. In the simulation, $m$ and $I$ are selected as 0.5 and 0.3, respectively.

Using a conventional PID controller, the simulated robot tracks the desired pose $q_d = [0.6, -0.6, 0.6, 0, 0, 0]^T$ from its initial position $[0, 0, 0, 0, 0, 0]$. We performed simulations with two proposed strategies, and the next two subsections detail the simulation results for each strategy.

*6.1. Simulation 1*

First, we constructed a simulation to validate the effectiveness of a tension distribution approach that satisfies the tension requirements and tracks the computed wrench without considering the optimization problem associated with tension distribution. The $\alpha$ in this method is $\alpha = 10$, while the control gains are $K_d = 10$, $D_d = 10$, $K_i = 0.1$. The desired pose is selected as $q_d = [0.6, -0.6, 0.6, 0, 0, 0]^T$. Figure 4 represents the simulation result of this method.

From Figure 4, it is evident that the tension of the cable falls between 0 N and 20 N. Additionally, the figure shows that the tension satisfies the tension constraints while preserving the tracking of the computed control wrench. Additionally, from these results it is found that the robot's tracking error is reduced to zero, as the desired pose is continually achieved. In the comparison of the tracking error in panel (c) of Figure 4, the identical overlap between the results of the original direct calculation using a computed wrench (represented by the gray line in the figure) and the results of the computation using our

approach (represented by the blue line in the figure) demonstrates the efficacy of our method.

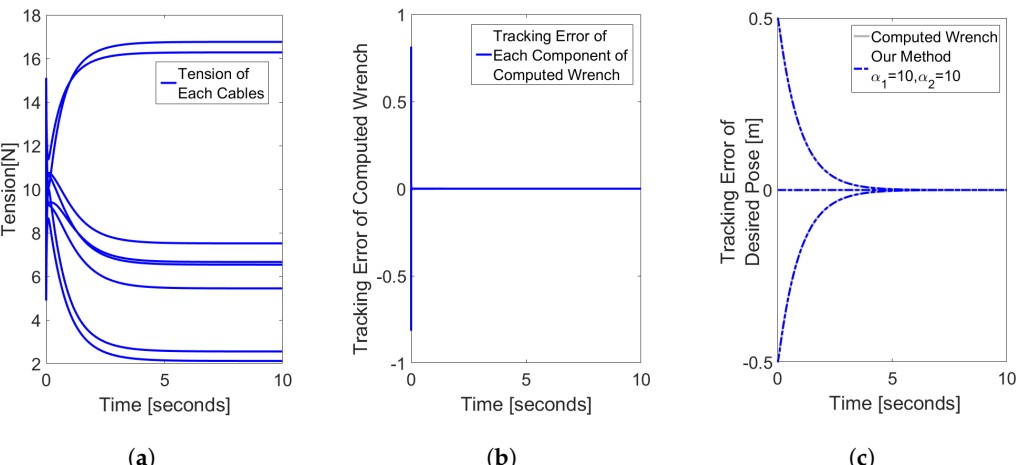

(**a**)      (**b**)      (**c**)

**Figure 4.** Results of Simulation 1. Parameters and desired pose are selected as $\alpha = 10$, $K_d = 10$, $D_d = 10$, $K_i = 0.1$, $q_d = [0.6, -0.6, 0.6, 0, 0, 0]^T$. (**a**) Tension distribution in Simulation 1. (**b**) Tracking error of computed wrench. (**c**) Tracking error of desired pose.

### 6.2. Simulation 2

Next, we conducted a simulation to validate the effectiveness of the second tension distribution method, taking into account the optimization problem of tension distribution while simultaneously satisfying tension limits and tracking the computed wrench. In this method, $\alpha_1$ and $\alpha_2$ were selected as $\alpha_1 = 15$ and $\alpha_2 = 12$, respectively, the control gain was selected as $K_d = 10$, $D_d = 10$, $K_i = 0.1$, and the desired pose was selected as $q_d = [0.6, -0.6, 0.6, 0, 0, 0]^T$. Figure 5 shows the simulation results of this method.

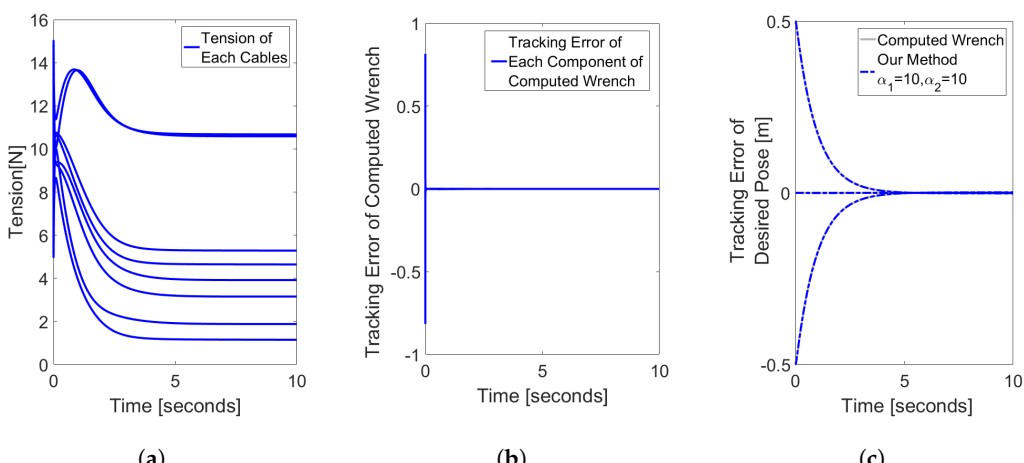

(**a**)      (**b**)      (**c**)

**Figure 5.** Results of Simulation 2. Parameters and desired pose are selected as $\alpha_1 = 15$ , $\alpha_2 = 12$, $K_d = 10$, $D_d = 10$, $K_i = 0.1$, $q_d = [0.6, -0.6, 0.6, 0, 0, 0]^T$. (**a**) Tension distribution in Simulation 2. (**b**) Tracking error ofcomputed wrench. (**c**) Tracking error of desired pose.

From Figure 5, using our second proposed tension distribution method with the added goal of lowering the tension vector's norm, it can be observed that all tensions decrease to values below 8 N. Neither the desired wrench tracking ability nor the tracking ability of the desired pose is diminished concurrently. Consequently, these results demonstrate that our proposed strategy is valuable and effective. Additionally, in panel (c) of Figure 5 it can be seen that the results of the original direct calculation using a computed wrench and the results of our method overlap perfectly, further indicating the effectiveness of our method.

*6.3. Simulation 3*

In this simulation, the gravity center of end-effector $P_G$ is selected as an uncertain parameter of the Jacobian matrix. We select the origin of $\mathbb{C}_o$ as the geometrical center of the end-effector and set the vector from this origin to the gravity center of end-effector as an unknown $a$. Both $\alpha_1$ and $\alpha_2$ are selected as $\alpha_1 = 0.1$ and $\alpha_2 = 0.1$, respectively. The desired pose is $q_d = [0.6, -0.6, 0.6, 0, 0, 0]^T$.

From Figure 6, it can be seen that the proposed tension distribution method is effective, as the tension satisfies the tension constraint and varies smoothly. From the results of the convergence errors of the desired wrench in Equation (11), shown in panel (b) of Figure 6, it is clear that the adaptive rule proposed in this paper is useful. Most importantly, by combining panels (a) of Figure 6, (b) of Figure 6, and (d) of Figure 6, it can be seen that the high-frequency fluctuation exerted in $\hat{a}$'s adaptive process has little impact on the tension variation when low parameters of $\alpha_1 = 0.1$ and $\alpha_2 = 0.1$ are selected. In addition, it can be seen from the results of the tracking error shown in panel (c) of Figure 6 that the robot's state converges to the desired position, which proves that the proposed hyperbolic tangent function-based desired wrench approximation method and the adaptive tuning law of Jacobian parameter allow a cable-driven robot with an unknown gravity center to successfully complete the position tracking task. This implies that our method can be used to implement this type of control method in a cable-driven robot, albeit with the drawback of a potentially high-fluctuation control wrench.

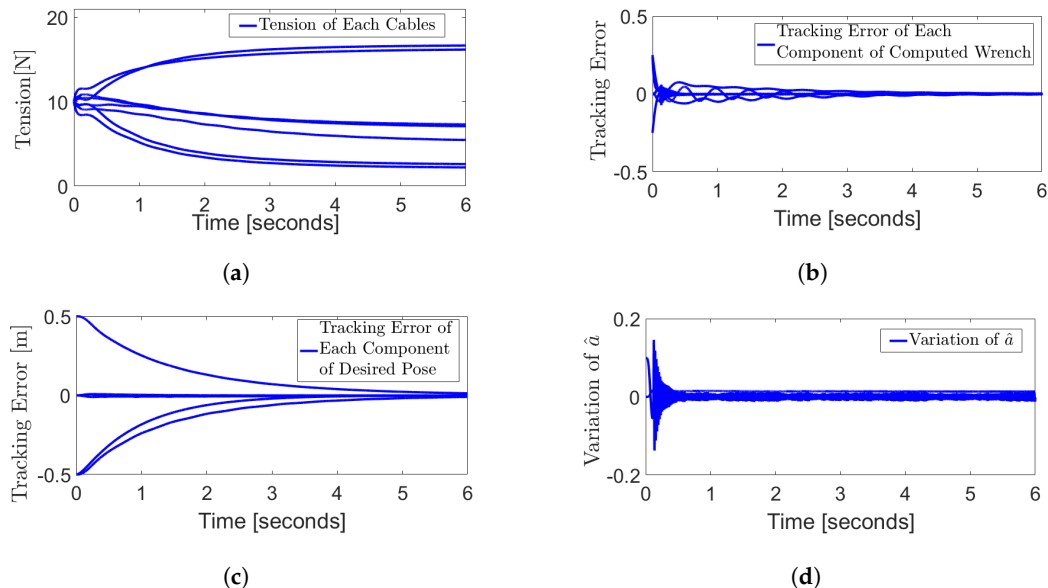

**(a)**

**(b)**

**(c)**

**(d)**

**Figure 6.** Results of Simulation 3. Parameters and desired pose are selected as $\alpha_1 = 15$, $\alpha_2 = 12$, $K_d = 0.1$, $D_d = 0.1$, $K_i = 0$, $\Gamma = 20$, $q_d = [0.6, -0.6, 0.6, 0, 0, 0]^T$. (**a**) Tension distribution. (**b**) Tracking error of computed wrench. (**c**) Tracking error of desired pose. (**d**) Variation of $\hat{a}$.

## 7. Discussion: Comparison with Other Tension Distribution Methods

*7.1. Computational Complexity and Calculation Speed*

Realizing real-time control of cable-driven robots requires tackling the problem of computational complexity. In each control loop of the suggested approach for tension distribution, it is necessary to compute Equations (33) and (34), the integral of $\hat{\dot{L}}_a \hat{\dot{S}}$, and Equation (28), which comprises the matrix dot product and summation. In addition, there exists no iterative calculation or optimization procedure in either control loop. Therefore, this approach has low enough computational complexity for use in real-time applications.

Here, we test the computational speed of the proposed algorithm in Matlab2016b with an AMD Ryzen 7 5800H CPU and 3.20 GHz Radeon GPU. Because the closed-form method is the fastest known method for distributing tension, we compared the calculation speed of our method with the closed-form method. We found that the closed-form method requires

24 μs for one control loop, whereas our method requires 35 μs. Although our approach is slightly slower than the closed-form method, our proposed algorithm's calculation speed is fast enough to process tension distribution calculations in real-time.

### 7.2. Continuous Solution

In practice, continuous solutions are essential. Although non-continuous tendon forces may be a feasible solution, they result in discontinuities in motor torques, which in turn produce vibrations and high mechanical loads. Many existing real-time tension distribution algorithms, such as the closed-form, improved closed-form, Barycentric, and puncture methods, attempt to solve this problem by analyzing the structure matrix $A^T$ and finding an appropriate continuous trajectory in the kernel space of $A^T$ to construct the continuously varying tension solution. The issue of the discontinuity that may be imposed in the calculated control wrench owing to white noise in the sensing process, discontinuous changes of the desired pose, buffering the influence of unexpected external loads, etc., has not yet been considered in the issue of tension distribution.

Our approach theoretically guarantees the continuity of the solution of tension distribution based on a previously selected tension function constructed by $tanh(:)$. Then, tracking the computed wrench based on the selected tension function naturally provides the continuously varying internal tension that previous research attempts to find in the kernel space of $A^T$ and allows for selection of the convergence rate of tracking for the computed wrench by adjusting the parameter $\alpha_1, \alpha_2$ to filter out any excessive discontinuous changes of the computed wrench that may exist, resulting in an acceptable continuous internal tension.

We compared our method with the closed-form method in the following two cases: 1. Measurement noise exists in $q$, and 2. The desired pose changes discontinuously at some time. Here, we selected closed-form in the comparison because it has the fastest calculation speed, provides an effective way to perform the tension distribution, and is a key part of other real-time algorithms such as the improved closed-form method and the puncture method. The improved closed-form method and the puncture method behave similarly when the control wrench is discontinuous.

For the case of existing white noise in sensing $q$, we compared our method to the closed-form method in Figure 7. Here, white noise is generated using the MatLab Simulink block 'Band-Limited White Noise', with the noise power selected as 0.0000001.

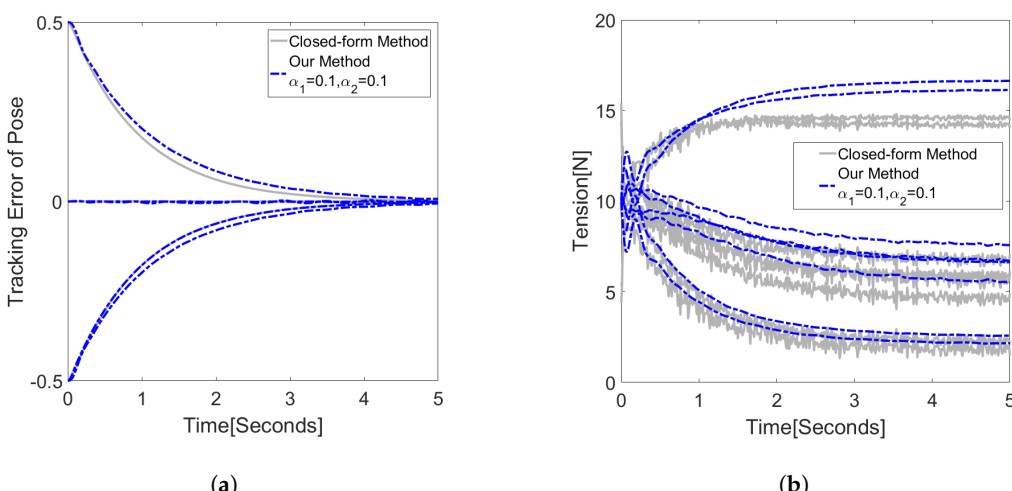

(**a**)                                        (**b**)

**Figure 7.** Comparison of our method and closed-form method (noisy case). Parameters and desired pose are selected as $\alpha_1 = 0.1$, $\alpha_2 = 0.1$, $K_d = 2$, $D_d = 1$, $K_i = 0$, $q_d = [0.6, -0.6, 0.6, 0, 0, 0]^T$. (**a**) Tracking error of pose. (**b**) Tension result.

As shown in Figure 7, the existence of white noise in the measurement results of $q$ and the fluctuation of the gray line in the tension results reflect the results of the closed-form

method, with a high-frequency vibration that can cause discontinuities in the torque of the motors. In contrast to this result, the blue line, which represents the result of our approach of selecting lower $\alpha_1 = 0.1$, and $\alpha_2 = 0.1$, varies smoothly, and the high vibration is eliminated. In addition, compared to the prior closed-form method, although convergence in tracking the desired pose of our method with the selection of $\alpha_1 = 0.1$, $\alpha_2 = 0.1$ is a little slower, the difference between the two lines is not significant, and the tracking errors converge to 0.

Moreover, we compared our method to the closed-form method in the case where the desired pose changes at some point. Here, the shift times of the desired pose are 5 s and 10 s.

The gray line of the closed-form method varies at 5 s and 10 s, changing to a maximum of roughly 5 N, as shown in Figure 8, and a significant disturbance in the tension occurs when the desired position is changed. The blue dotted line in the figure shows that our method with the selection of $\alpha_1 = 0.1$ and $\alpha_2 = 0.1$ efficiently reduces this disturbance and eliminates the impact of the discontinuous shift of the desired pose. The tracking error of the desired pose of the two methods shows that our method can effectively track the desired pose even with a low $\alpha_1 = 0.1$, $\alpha_2 = 0.1$ in this case; however, it is slightly slower than the closed-form method.

Thus, when the results of these two cases are compared, it can be seen that our method is better able to handle the discontinuity in the computed control wrench and keep the tension change from being interrupted.

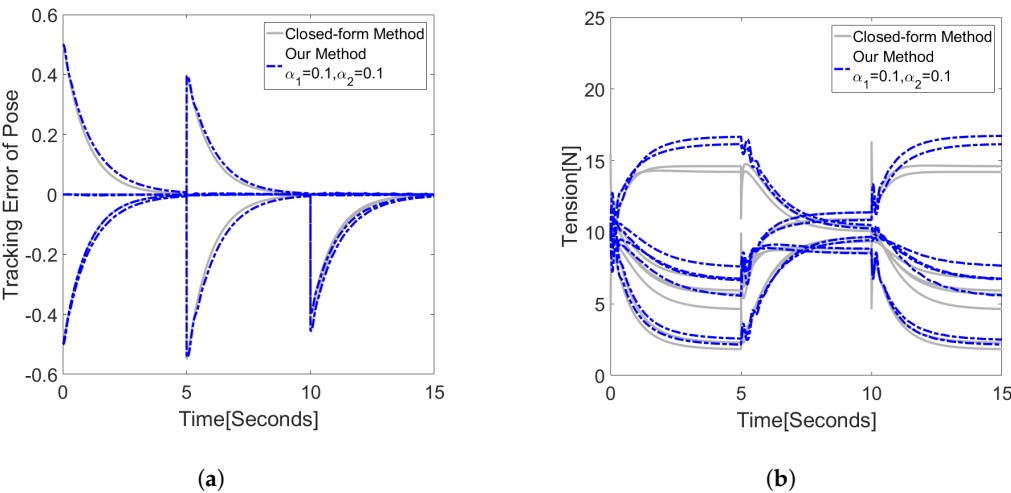

|            (a)            |            (b)            |

**Figure 8.** Comparison of our method and closed-form method (shifting desired pose). Parameters and desired pose are selected as $\alpha_1 = 0.1$, $\alpha_2 = 0.1$, $K_d = 2$, $D_d = 1$, $K_i = 0$, $q_d = [0.5, -0.5, 0.5, 0, 0, 0]^T$ ($0\,\mathrm{s} \rightarrow 5\,\mathrm{s}$), $q_d = [0.1, 0.05, 0.5, 0, 0, 0]^T$ ($5\,\mathrm{s} \rightarrow 10\,\mathrm{s}$) and $q_d = [0.5, 0.5, 0.5, 0, 0, 0]^T$ ($10\,\mathrm{s} \rightarrow 15\,\mathrm{s}$). (**a**) Tracking error of pose. (**b**) Tension result.

### 7.3. Force Level and Force Level Changing

The existing tension distribution method may attempt to select the minimum, maximum, average, or any solution of the force level of tension (p99, [18]). In fact, the force level of the cables affects the stiffness of the cable-driven robot [18]. Especially in applications where robots may contact the external environment, the issue of the variable stiffness control of the robot is crucial for completing such sophisticated tasks. As a result, a tension distribution algorithm that is capable of realizing any tension distribution is required, rather than just focusing on achieving a specific minimum or maximum force level.

In our method, we select $f_0$ in Equation (27) as

$$f_0 = A^{+T}\tau_d + (\boldsymbol{U}_{m \times m} - A^{+T}A^T)s \tag{61}$$

where $s \in R^m$ denotes an optional vector. Here, $f_0$ represents the point in the tension's solution space satisfying $A^T f = \tau_d$ that has the minimum distance with vector $s$. Therefore, by changing the vector $s$, selecting $f_0$ in this equation, and using our method, we can obtain an appropriate force level. Notably, when setting the $s$ as zero (as in Equation (27)), we can obtain the point in the tension's solution space satisfying $A^T f = \tau_d$ that is the closest to the origin. Additionally, this point is necessary for the puncture method.

The example below shows the ability of our method in terms of selecting the force level and performing force-level changes. Here, $s$ is selected as $s = [10, \ldots, 10]^T \in R^8$ from 0 s to 5 s and $s = [15, \ldots, 15]^T \in R^8$ is selected from 5 s to 15 s.

From Figure 9, it can be seen that the gray line in the tension results represents the $f_0$ relating to the selected $s$ and the blue line represents the tension distribution result of our method. From 0 s to 5 s, all elements of $f_0$ satisfy tension constraints, and the tension calculated by our method can precisely track $f_0$. After 5 s, with increasing $s$, elements of $f_0$ become bigger than 20 Nm which does not satisfy the tension constraint. In this situation, the tensions calculated using our method converge to an appropriate force distribution, which has the closest distance to $f_0$ in the feasible solutionsm and the force level is enlarged to an extent. Notably, when the force level is changed at 5 s, our method yields satisfactory results for the continuous variation of tension.

These results demonstrate that our method can be used to achieve any required feasible force level and the change in force level during the robot's movement while maintaining the continuous variation of tension.

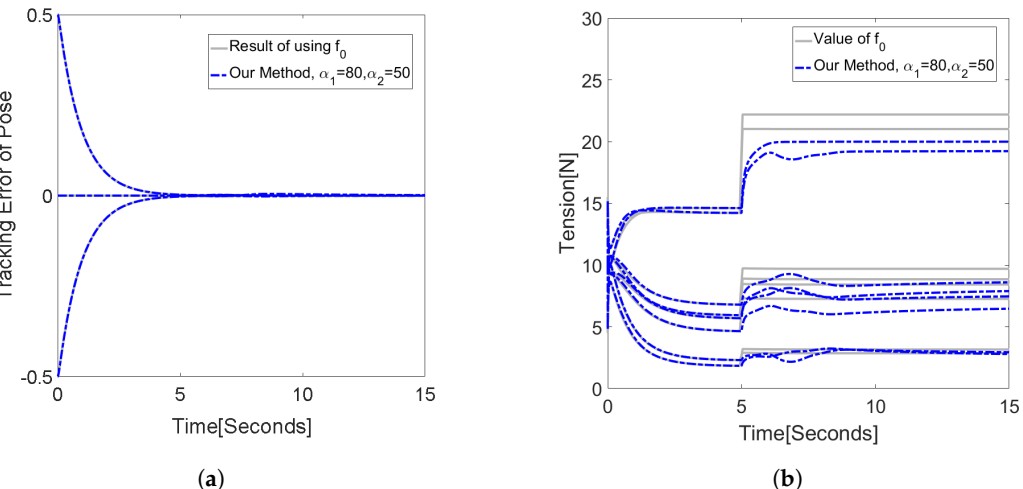

**Figure 9.** Force level changing with our method. Parameters and desired pose are selected as $\alpha_1 = 80$, $\alpha_2 = 50$, $K_d = 2$, $D_d = 1$, $K_i = 0$, $q_d = [0.6, -0.6, 0.6, 0, 0, 0]^T$, $s = [10, \ldots, 10]^T \in R^8$ (0 s $\rightarrow$ 5 s) and $s = [15, \ldots, 15]^T \in R^8$ (5 s $\rightarrow$ 15 s). (**a**) Tracking error of pose. (**b**) Tension result.

## 8. Conclusions

In order to tackle the control problem of an over-constrained cable-driven robot with tension (control input) constraints and cable redundancy, we suggest a desired computed wrench method based on a hyperbolic function. In this method, a generated continuous function containing a hyperbolic tangent function is utilized to form the tension, and the variable of this function can be chosen to satisfy any conditions. This method is effective in solving the control problem of cable-driven robots due to its ability to ensure the continuity of cable tension and its straightforward calculation procedure. In addition, we propose a method for achieving optimal tension distribution objectives, such as the minimization of cable tension norms for energy saving. According to the simulation results, our two proposed methods are effective. By comparing our method with previous real-time tension distribution methods, the advantages of handling discontinuity with our method can be seen in terms of the computed wrench, force level changes, etc.

In future work, we intend to study the selection laws for the tracking parameters $\alpha_1$ and $\alpha_2$ in order to determine the best tuning laws. In addition to applying this technique in simulations, we want to utilize it in real-world scenarios involving cable-driven robots in order to perform tasks such as collision tackling, handling the cable strain and elasticity problem in the tension control, and managing the problems of aerial cable-towed robots, etc.

**Author Contributions:** Conceptualization, S.C.; Methodology, S.C.; Software, C.Q.; Writing—original draft, S.C.; Writing—review & editing, S.C.; Project administration, Z.L. and C.Q. All authors have read and agreed to the published version of the manuscript.

**Funding:** This research received no external funding.

**Institutional Review Board Statement:** Not applicable.

**Informed Consent Statement:** Not applicable.

**Data Availability Statement:** Not applicable.

**Conflicts of Interest:** The authors declare no conflict of interest.

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
