# Peer review of "Real-Time Tension Distribution Design for Cable-Driven Parallel Robot"

_applsci, doi:10.3390/app13010010_

Round 1
Reviewer 1 Report
Paper: Real-time Tension Distribution Design for Cable-Driven Robot
This paper proposes a computed wrench approaching method based on a hyperbolic function in order to tackle the control problem of an over-constrained cable-driven robot with tension (control input) constraints and cable redundancy. The work is interesting; however, the authors should do all of the following comments to improve the paper and to be in a suitable form.
Comments:
P1: More deep information about the methodology and the results should be included in the abstract.
P2: The literature review presented in the introduction part is weak. It should be improved and discussed widely with more references. In addition, the advantages and disadvantages of each method should be stated.
P3: At the end of the introduction part and after stating the main contribution of the paper, the outline of the paper should be presented.
P4: The references/sources of the most presented equations should be added. This is important.
P5: How Equation (12) is obtained? More details are required.
P6: It should be clear in the text why the Hyperbolic Tangent Function is used and its benefit.
P7: Block diagram, pseudocode, or flowchart should be added in the paper to show the presented equations. This can help for more understanding.
P8: What are the values of the PD controller gains? And how these values are obtained?
P9: It is better to add also figures in the paper to show the actual and the desired positions of the robot.
P10: The resolution of the figures is bad. It should be improved and also the writing on the figures should be clear and large.
P11: Comparing with other previous related published works should be included at the end of the paper particularly the tracking error. The following paper can also be used in comparison:
[Dynamics and Computed-Torque Control of a 2-DOF manipulator: Mathematical Analysis. International Journal of Advanced Science and Technology, 2019, 28 (12), pp.201-212. https://hal.archives-ouvertes.fr/hal-03598924 ]
P12: Some future work should be added at the end of conclusion.
P13: Finally, the English of the paper should be revised carefully as there are many errors.
Reviewer 2 Report
Please see attached PDF.

Round 2
Reviewer 1 Report
I have no other comments.
Author Response
Dear Reviewer 1,
Thank you for your review of our paper. We appreciate your feedback and are grateful for the opportunity to improve our work.
We are pleased to see that you have evaluated the English language and style in our paper as fine, with only minor spell check required. We have carefully proofread our paper and corrected any spelling errors we found.
We appreciate your thorough review and are grateful for your time and effort in evaluating our work. Thank you again for your valuable input.
Sincerely,
Sheng Cao
Reviewer 2 Report
Please see attached PDF.

Author Response
Dear Reviewer 2,
Thank you for your review of our paper and for your feedback on our manuscript's English language and style. We appreciate your comments and have made the necessary revisions to address your concerns.
We also appreciate your detailed feedback on the content of our paper and have included responses to your specific comments in the attached file. We hope that our answers address your concerns and address any questions you may have about our work.
Thank you again for your valuable input. We are grateful for the opportunity to improve our manuscript and hope you will consider our revised submission for publication.
Sincerely,
Sheng Cao
